# Decoupling Shared and Modality-Specific Subspaces in Multimodal Learning via Low-Rank Representation Fine-Tuning

## Abstract

Multimodal data in machine learning promises to improve generalization and performance on complex tasks. However, training multimodal models requires extensive paired datasets, can be computationally expensive, and lacks transparency by entangling shared and modality-specific signals in ways that hinder interpretability and control. In this work, we introduce MultiLoReFT: a low-rank representation fine-tuning framework for multimodal learning using pretrained unimodal models. Our approach extends low-rank representation finetuning to the multimodal setting and learns interpretable projection subspaces that decouple shared and modality-specific information. MultiLoReFT adaptively learns the rank of each subspace to best capture complementary contributions of each modality with minimal trainable parameters. Our method offers an efficient and scalable solution to adapting pretrained representations for multimodal reasoning, enabling interpretable fine-tuning across both synthetic and real-world benchmarks.

## 1 Introduction

The growth of multimodal data, ranging from image-caption pairs to multimodal diagnostic data, has enabled a wide range of applications (Liang et al., 2024) in vision-and-language modeling (Chen et al., 2024; Sun et al., 2025), medical diagnostics (Steyaert et al., 2023; Zhou et al., 2023), and biology (Cui et al., 2025). These applications require learning effective joint representations that capture both the common semantics across modalities and the unique information each modality provides. However, training multimodal models from scratch typically demands large amounts of aligned, high-quality multimodal data, which are often scarce or expensive to obtain in real-world settings (Baltrušaitis et al., 2019). This has motivated recent approaches to reuse powerful unimodal encoders pretrained on large-scale corpora, and adapt them for multimodal tasks through fine-tuning (Kim & Kim, 2024; Miyazawa et al., 2022).

While fine-tuning enables flexible reuse of pretrained models, it can be computationally intensive and parameter inefficient. Recent work has introduced parameter-efficient fine-tuning strategies such as low-rank adaptation (LoRA) (Hu et al., 2021) and low-rank representation fine-tuning (LoReFT) (Wu et al., 2024) that directly update internal representations in low-dimensional subspaces instead of fine-tuning the model parameters. These approaches achieve comparable performance to full fine-tuning with less computation. They are particularly attractive in data-limited regimes, like multimodal cohorts, as they reduce the risk of overfitting while preserving pretrained knowledge.

In this work, we extend low-rank representation fine-tuning to multimodal representation learning. We propose MultiLoReFT, a framework that efficiently fuses information from multiple pretrained unimodal encoders while simultaneously disentangling shared and unique information from each modality. This decoupling increases interpretability by providing insights into the relative contributions of each modality (Tsai et al., 2019), improves generalization across domains, and allows for better handling of missing modalities. MultiLoReFT offers a self-supervised solution to augment unimodal representations with cross-modal information that generalize to any downstream label. It learns structured low-rank projection matrices that define orthogonal shared and modality-specific subspaces to decouple the unique information contribution of each modality. Leveraging the structure of each subspace, we incorporate a novel adaptive pruning strategy that enables the model to

dynamically reduce the rank of each projection matrix. This results in learning the amount of information each subspace contains while improving efficiency by avoiding over-parameterization. We evaluate our approach on synthetic and real-world multimodal datasets, demonstrating its ability to successfully fuse multimodal representations, learn shared and modality-specific subspaces that identify and decouple the unique and shared information, and allocate representational capacity effectively for each subspace. By bridging multimodal representation disentanglement and efficient fine-tuning, our method offers a principled and lightweight approach to leveraging pretrained unimodal models in multimodal scenarios. This lays the groundwork for interpretable and adaptable multimodal systems in data-scarce settings where training multimodal models can be challenging.

## 2 RELATED WORK

**Multimodal Representation Learning.** A central challenge in multimodal learning is how to integrate heterogeneous signals into effective representations (Liang et al., 2021). Early approaches rely on simple early, intermediate or late fusion mechanisms (Boulahia et al., 2021). Coordinated representation approaches align unimodal encoders into a shared embedding space, often with contrastive or retrieval-based objectives (Radford et al., 2021; Hager et al., 2023). Fusion-based models remain widely used, ranging from simple concatenation or pooling strategies (Baltrušaitis et al., 2019) to attention-based architectures that explicitly capture cross-modal interactions (Tsai et al., 2019; Jayakumar et al., 2020). Recent studies have taken a more analytical view, quantifying redundancy and complementarity information between modalities using Partial Information Decomposition (PID) (Liang et al., 2023a; Zhang et al., 2025). These insights emphasize that naive fusion lack clarity on the structure of modality-specific and shared information, motivating the development of disentanglement frameworks.

**Disentangled Multimodal Representations.** A complementary line of work aims to explicitly separate shared and modality-specific information in multimodal settings. These methods often define information components with respect to downstream tasks; for instance, FactorCL (Liang et al., 2023b) aligns modality-invariant features with supervision signals while preserving unique factors using a factorized contrastive learning. Triple Disentanglement (Zhou et al., 2025) further decomposes representations into shared, relevant, and irrelevant modality-specific components using a transformer-based encoder–fusion design. Other approaches offer self-supervised alternatives to information decomposition. DRIM-U (Robinet et al., 2024) enforces disentanglement through reconstruction and adversarial regularization, while APOLLO (Zhang et al., 2024b) leverages latent optimization to learn partially shared embeddings that generalize through trained encoders. These approaches move beyond fusion to provide a more structured account of modality interactions.

**Multimodal Fine-tuning.** The difficulty of collecting large-scale paired multimodal datasets has motivated research on leveraging unimodal models for multimodal learning (Zhang et al., 2024a). Existing approaches range from fusion of unimodal encoders (Miyazawa et al., 2022; Norelli et al., 2023) to direct fine-tuning for multimodal tasks (Zhai et al., 2022). Yet, fine-tuning large models remains challenging, particularly when available multimodal cohorts are small (Vieira et al., 2024). In language models, representation-level fine-tuning rather than full model adaptation has shown strong effectiveness for downstream tasks (Wu et al., 2024; Hu et al., 2021). Extensions to multimodal learning (Liu et al., 2025) similarly demonstrate gains in both performance and flexibility. Building on this, our method introduces a multimodal fine-tuning approach that improves task performance and clarifies the division of information between shared and unique components.

## 3 MULTIMODAL REPRESENTATION FINETUNING (MULTILOREFT)

We introduce a low-rank representation fine-tuning framework for multimodal learning called MultiLoReFT that decomposes pretrained unimodal representations into shared and modality-specific components. As shown in Figure 1, our method operates on top of frozen pretrained encoders, requiring only a small number of additional parameters. This design enables efficient multimodal fusion while adding interpretability by explicitly disentangling modality-specific contributions from shared information.

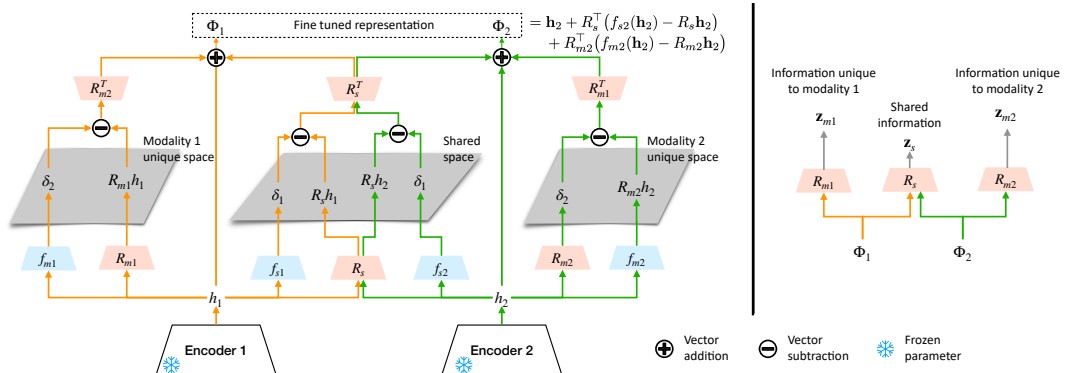

Figure 1: Overview of **MultiLoReFT**. Pretrained unimodal encoders (with frozen parameters) produce representations $h_1$ and $h_2$, which are fine-tuned through low-rank projections into shared ($R_s$) and modality-specific ($R_{m1}, R_{m2}$) subspaces. Nonlinear transforms ($f_s, f_m$) learn the representation edit ($\delta$) in the lower-rank space that needs to be subtracted from the projection of the representations. Once edited in the low-rank space, the representations are projected back to form the fine-tuned representations $\Phi_1$ and $\Phi_2$. The right panel illustrates the decoupling of information into shared ($\mathbf{z}_s$) and modality-unique ($\mathbf{z}_{m1}, \mathbf{z}_{m2}$) components.

### 3.1 REPRESENTATION FINE-TUNING WITH MULTILOREFT

We build on representation fine-tuning (Wu et al., 2024) to adapt pretrained unimodal encoders for multimodal learning. The key idea is to apply structured, low-rank interventions on pretrained representations, steering them toward disentangled multimodal subspaces that capture both shared and modality-specific factors. Consider two pretrained unimodal encoders $E_1$ and $E_2$ for modalities 1 and 2. Given inputs $x_1$ and $x_2$, the encoders produce representations $\mathbf{h}_1, \mathbf{h}_2 \in \mathbb{R}^d$. We learn fine-tuned representations $\Phi_1, \Phi_2 \in \mathbb{R}^d$ that (i) preserve the expressive power of the representations, (ii) capture multimodal interactions, and (iii) disentangle shared and modality-specific information.

We introduce three low-rank subspaces: one shared across modalities and two modality-specific. Projection matrices $R_s, R_{m1}, R_{m2} \in \mathbb{R}^{r \times d}$ map fine-tuned representations into these subspaces, yielding the components in Eq. 1. In this formulation, $\mathbf{z}s$ represents shared information, while $\mathbf{z}_{m1}$ and $\mathbf{z}_{m2}$ capture information unique to each modality, providing interpretable building blocks for downstream multimodal tasks.

$$\mathbf{z}_s = R_s \Phi_1 = R_s \Phi_2, \quad \mathbf{z}_{m1} = R_{m1} \Phi_1, \quad \mathbf{z}_{m2} = R_{m2} \Phi_2. \tag{1}$$

Instead of finetuning encoders, MultiLoReFT learns lightweight transformations $f_{si}, f_{mi}$ in the low-rank subspaces. The fine-tuned representation of each modality $i \in \{1, 2\}$ is computed as:

$$\Phi_1 = \mathbf{h}_1 + R_s^\top \big(f_{s1}(\mathbf{h}_1) - R_s \mathbf{h}_1\big) + R_{m1}^\top \big(f_{m1}(\mathbf{h}_1) - R_{m1} \mathbf{h}_1\big), \tag{2}$$

$$\Phi_2 = \mathbf{h}_2 + R_s^\top \big(f_{s2}(\mathbf{h}_2) - R_s \mathbf{h}_2\big) + R_{m2}^\top \big(f_{m2}(\mathbf{h}_2) - R_{m2} \mathbf{h}_2\big). \tag{3}$$

These edits operate only in the low-rank subspaces, making training efficient while ensuring interpretability. The shared subspace isolates common information, and the modality-specific subspaces capture unique signals. The transformations learn how to update the representations in each space.

### 3.2 OBJECTIVE

Unlike prior ReFT methods, fine-tuning in our framework is not guided by a supervised task objective but instead by structural constraints imposed on the representation space. The goal is to adapt pretrained unimodal embeddings so that their projections into learned subspaces exhibit disentanglement while still enabling effective multimodal fusion. To this end, we optimize a composite objective that encourages (i) independence between shared and modality-specific components, (ii) orthogonality between subspaces, and (iii) preservation of information from the original unimodal embeddings. The overall loss is composed of 3 componenets:

$$\mathcal{L} = \lambda_1 \mathcal{L}_{\text{indep}} + \lambda_2 \mathcal{L}_{\text{orth}} + \lambda_3 \mathcal{L}_{\text{MI}}. \tag{4}$$

**Independence loss.** ($\mathcal{L}_{\textbf{indep}}$)  To ensure that shared and modality-specific components capture complementary information, we minimize their statistical dependence using the Hilbert–Schmidt Independence Criterion (HSIC) (Gretton et al., 2007). HSIC is a nonparametric measure of statistical dependence between two random variables. Given random variables $X$ and $Y$ with kernels $k$ and $l$, when the kernels are characteristic (e.g., Gaussian RBF or Laplace), $\text{HSIC}(X, Y) = 0$ if and only if $X$ and $Y$ are statistically independent. Independence is enforced not only between the shared and private subspaces of each modality but also between the two modality-specific subspaces. This prevents leakage of redundant shared information into the private components:

$$\mathcal{L}_{\text{indep}} = \text{HSIC}(\mathbf{z}_{s1}, \mathbf{z}_{m1}) + \text{HSIC}(\mathbf{z}_{s2}, \mathbf{z}_{m2}) + \text{HSIC}(\mathbf{z}_{m1}, \mathbf{z}_{m2}). \tag{5}$$

MultiLoReFT uses an empirical, unbiased estimator of H that can be minimized during training to enforces nonlinear independence between the representation components as $\text{HSIC}(X, Y) = \frac{1}{(n-1)^2} \text{tr}(KL)$, where $K, L \in \mathbb{R}^{n \times n}$ are centered kernel matrices. We use an RBF kernel in all experiments.

**Orthogonality loss.** ($\mathcal{L}_{\textbf{orth}}$)  While independence ensures statistical separation, we further enforce disjointness between the shared subspace $R_s$ and modality-specific subspaces $R_{m1}$ and $R_{m2}$. This is done by minimizing the Frobenius norm of their pairwise inner products:

$$\mathcal{L}_{\text{orth}} = \|R_s R_{m1}^\top\|_F + \|R_s R_{m2}^\top\|_F. \tag{6}$$

This constraint strengthens disentanglement by ensuring the subspaces are orthogonal. While HSIC guarantees that subspaces do not carry redundant information, orthogonality ensures that their basis vectors do not overlap in representation space. For instance, two statistically independent variables could still align geometrically (colinear bases), and two orthogonal directions could still exhibit nonlinear dependence.

**Cross-modal mutual information loss.** ($\mathcal{L}_{\textbf{MI}}$)  To ensure that the fine-tuned projections retain information from the original unimodal embeddings $\mathbf{h}$, we adopt an InfoNCE-style contrastive loss as shown in Equation 7. InfoNCE provides a lower bound on the true mutual information, and maximizing this bound reserves high mutual information between the projections and their sources. Therefore, the disentangled shared and modality-specific projections remain *sufficient summaries* of their original embeddings.

$$\mathcal{L}_{\text{MI}} = -\frac{1}{2} \sum_{i=1}^{2} \log \frac{\exp\left(\langle \mathbf{h}_i, \mathbf{z}^{(i)} \rangle / \tau\right)}{\sum_{j=1}^{N} \exp\left(\langle \mathbf{h}_i, \mathbf{z}^{(j)} \rangle / \tau\right)}. \tag{7}$$

Where $\tau$ is the temperature parameter that controls the sharpness of the similarity distribution inside the softmax, $h_i$ is the pretrained embedding of modality $i$, and $N$ is the batch size. Here, $\mathbf{z}^{(i)}$ is formed by concatenating the modality-specific projection from modality $i$ with the shared projection from the opposite modality. This design enforces consistency of shared components across modalities, ensuring that they encode modality-agnostic information.

To avoid hand-tuning regularization weights $\lambda$, we adopt *Gradient Normalization* (Chen et al., 2018), which balances the contributions of each objective by equalizing their gradient magnitudes. We demonstrate the contribution of each loss component to successful representation learning through an ablation study presented in the Appendix A.2.2.

## 3.3 TRAINING

### 3.3.1 TRAINING PROCEDURE

We adopt a multi-stage training strategy that progressively learns different components of the model, as outlined in Algorithm 1 and described through the following steps.

- **Stage 1 (Shared).** We first optimize only the shared subspace $R_s$ and associated parameters to capture cross-modal information that is maximally aligned between the two modalities. At this stage, training is driven solely by the mutual information loss $\mathcal{L}_{\text{MI}}$.

- **Stage 2 (Private).** Next, we optimize the modality-specific subspaces $R_{m1}$ and $R_{m2}$ to extract private components that encode complementary information to the learned shared while remaining independent of the shared space.

- **Stage 3 (Joint).** Finally, we fine-tune all parameters jointly and initiate the adaptive pruning process, allowing the model to refine both shared and modality-specific representations.

Transitions between stages are determined adaptively using a validation-based convergence criterion. Specifically, we monitor the validation loss and trigger a stage switch when (i) its relative improvement falls below a minimum threshold, and (ii) this condition persists for a number of consecutive epochs specified by the patience parameter. In all our experiments, we use a minimum relative improvement of $0.001$ within a patience window of 40 epochs (increased to 100 epochs for the joint stage, to account for recovery after pruning). Appendix A.2.2 provides ablation results to show the importance of staged learning and the pruning procedure.

---

**Algorithm 1:** MultiLoReFT Training

---

**Input:** Multimodal datasets $(\mathcal{D}_{\text{train}}, \mathcal{D}_{\text{val}})$; pretrained encoders $(E_1, E_2)$; pruning threshold $\epsilon$
; Convergence criteria
**Variables:** Projection matrices $R_s, R_{m1}, R_{m2}$; transform functions $f_{s1}, f_{s2}, f_{m1}, f_{m2}$

---

`trainable_stage` ← "shared"
**while** not converged **do**
    **foreach** *minibatch* $(\mathbf{x}_1, \mathbf{x}_2)$ *in* $\mathcal{D}_{\text{train}}$ **do**
        $(\mathbf{h}_1, \mathbf{h}_2) \leftarrow E_1(\mathbf{x}_1), E_2(\mathbf{x}_2)$
        $\Phi_1 = \mathbf{h}_1 + R_s^\top(f_s(\mathbf{h}_1) - R_s\mathbf{h}_1) + R_{m1}^\top(f_m(\mathbf{h}_1) - R_{m1}\mathbf{h}_1)$
        $\Phi_2 = \mathbf{h}_2 + R_s^\top(f_s(\mathbf{h}_2) - R_s\mathbf{h}_2) + R_{m2}^\top(f_m(\mathbf{h}_2) - R_{m2}\mathbf{h}_2)$
        $\mathbf{z}_{s1} \leftarrow R_s\Phi_1$;    $\mathbf{z}_{m1} \leftarrow R_{m1}\Phi_1$;
        $\mathbf{z}_{s2} \leftarrow R_s\Phi_2$;    $\mathbf{z}_{m2} \leftarrow R_{m2}\Phi_2$;
        **if** `trainable_stage` = shared **then**
            $\mathcal{L} \leftarrow$ GradNorm$([L_{\text{mi}}])$
            **Train:** $R_s, f_{s1}, f_{s2}$
        **else if** `trainable_stage` = private **then**
            $\mathcal{L} \leftarrow$ GradNorm$([L_{\text{orth}}, L_{\text{ind}}, L_{\text{mi}}])$
            **Train:** $R_{m1}, R_{m2}, f_{m1}, f_{m2}$
        **else** joint
            $\mathcal{L} \leftarrow$ GradNorm$([L_{\text{orth}}, L_{\text{ind}}, L_{\text{mi}}])$
            **Train:** $R_s, f_{s1}, f_{s2}, R_{m1}, R_{m2}, f_{m1}, f_{m2}$

    `// Validation, pruning, and stage control`
    Evaluate validation losses $\mathcal{L}_{\text{val}}$ on $\mathcal{D}_{\text{val}}$
    **if** *stage is joint **and** MI within* $10\%$ *of best MI* **then**
        Adaptive Rank pruning$(\epsilon)$ (Alg. 2)
    **if** Convergence criteria is met w.r.t $\mathcal{L}_{val}$ **then**
        advance stage: shared → private → joint

---

### 3.4 RANK ADAPTATION VIA PRUNING

A key challenge in disentangled representation learning is determining the dimensionality of shared and modality-specific subspaces: fixing ranks a priori risks underfitting when too small, or redundancy and leakage when too large. To address this, we adopt a dynamic rank adaptation mechanism that prunes low-energy directions. During training, we compute the singular value decomposition (SVD) of each projection matrix $R_s, R_{m1}, R_{m2}$ as $R = USV^\top$, with singular values $S = \text{diag}(\sigma_1, \dots, \sigma_r)$. Dimensions with $\sigma_i$ below a threshold $\epsilon$ are pruned, and the matrices are updated with a rotated, compressed basis $\tilde{R} = \text{diag}(S_{1:k})V_{1:k}^\top$, ensuring orthogonality and alignment with dominant directions (Algorithm 2). This rank adaptation improves robustness and eliminates the need for manual rank tuning, as further shown in Appendix A.2.2.

---

**Algorithm 2:** Adaptive Rank Pruning

---

**Input:** Matrices $R_s, R_{m1}, R_{m2}$; threshold $\epsilon$

**foreach** $R \in \{R_s, R_{m1}, R_{m2}\}$ **do**
    $R = U \operatorname{diag}(S) V^\top$
    Find projected ranks $k$ based on $S_i < \epsilon$
    Select $k$ by pruning singular values below $\epsilon$ (clip to keep at least one)
    Form $\tilde{R} \leftarrow \operatorname{diag}(S_{1:k}) V_{1:k}^\top$;
    For each associated $f$, replace last Linear layer to output $k$ and rotate weights by $U_{:,1:k}^\top$
    $R \leftarrow \tilde{R}$
Refresh optimizer param groups

---

## 4 EXPERIMENTS

We evaluate MultiLoReFT on both simulated and real-world datasets. Our experiments are designed to answer two key questions: (1) Does the model effectively decouple shared and modality-specific information, and how is modality-relevant information distributed across these components? (2) Does fine-tuning improve multimodal representations for downstream tasks by capturing joint information more effectively?

### 4.1 DATASETS AND BASELINES

We first conduct controlled evaluations on simulated datasets, where the ground-truth generative structure is known. Each dataset includes conditional, joint, and unique labels, enabling targeted validation of different aspects of multimodal representation learning. We then scale to large, real-world datasets to assess the applicability of our method with pretrained encoders.

- Simulation I & II. Two synthetic multimodal datasets with controlled generative processes. Simulation I is constructed from independent latent factors drawn from diverse distributions, while Simulation II introduces dependencies across some factors, creating correlated shared and unique components. Both datasets provide labels that are modality-specific, shared, and conditional, enabling systematic evaluation of disentanglement. Full details in Appendix A.1.

- Flickr30K-Multi (Elliott et al., 2016). A multilingual extension of the Flickr30K dataset containing image–caption pairs in five languages. Each image is paired with one caption per language, making language a modality-specific factor while semantic content remains shared across modalities. We use English and French captions for our experiments.

- Crema-D (Cao et al., 2014). An audio–visual dataset of multimodal emotion expression and perception, comprising 7,442 clips from 91 actors across diverse demographics. Each clip contains both facial and vocal expressions of fixed sentences. The dataset provides both shared emotional signals and modality-specific cues, along with contextual metadata (e.g., age, ethnicity).

We use established pretrained models for each modality: DINO Vision Transformer for images (Caron et al., 2021), BERT-base for English text (Devlin et al., 2019), LaBSE for multilingual text (Feng et al., 2022), Wav2Vec 2.0 base pretrained on Librispeech-960h for audio (Baevski et al., 2020), and a 3D ResNet-18 video encoder pretrained on Kinetics-400 (He et al., 2016; Kay et al., 2017). We compare our method against a broad set of multimodal representation learning frameworks, grouped into two categories:

- General fusion approaches. These methods integrate multimodal signals through concatenation, attention, or interaction mechanisms. As baselines, we use late fusion (Baltrušaitis et al., 2019), which simply concatenates modality representations, and an attention-based fusion model, which projects each modality into a common space and applies a lightweight self-attention layer to let the two embeddings interact before pooling into a fused representation. We also evaluate multiplicative interactions (**MI**) (Jayakumar et al., 2020), which extend tensor product fusion with learnable parameters to capture higher-order dependencies, and **contrastive learning**, which encourage aligned representations by pulling paired modalities closer and unpaired samples further.

- Decoupling approaches. These methods explicitly separate shared and modality-specific information. We consider **APOLLO** (Zhang et al., 2024b), an autoencoder-based model that decouples shared and unique components through latent optimization, directly learning embeddings for training samples before training encoders to generalize. We also benchmark **DRIM-U** (Robinet et al., 2024), which disentangles multimodal representations using three complementary objectives: enforcing similarity across shared embeddings, ensuring reconstruction fidelity, and adversarially regularizing unique modality-specific components. Since the original DRIM-U relies on task labels, we adopt its self-supervised variant presented in the paper, making it more comparable to our setting.

For fairness, all baselines are trained on the same pretrained unimodal embeddings as input. The only architectural differences lie in the method-specific adapters—for example, DRIM-U uses a discriminator-based decoupling module, whereas APOLLO learns latent parameters directly.

## 5 RESULTS

In this section, we present two sets of evaluations: (i) demonstrating that MultiLoReFT learns decoupled representations that correctly encode shared and modality-specific information, and (ii) showing that these decoupled representations also serve as strong multimodal features that improve downstream prediction tasks by leveraging cross-modal learning during fine-tuning.

### 5.1 DECOUPLING SHARED AND MODALITY-SPECIFIC INFORMATION

The objective of disentangling shared and modality-specific signals is to assess the unique and common information that each modality contributes in a multimodal setting. We compare MultiLoReFT against benchmark methods designed for disentanglement, and evaluate how the extracted components predict labels tied either to modality-specific or shared generative factors.

On the simulated datasets, we have labels corresponding to one of the underlying generative factors (Shared, M1, and M2). We measure the predictability of each representation component ($\mathbf{z}_s$, $\mathbf{z}_{m1}$, $\mathbf{z}_{m2}$) for these labels to test whether 1) the information is embedded in the right component, and 2) how well it is removed from the other components. Table 1 summarizes these results, measured as the performance of a logistic regression model trained on different representation components to predict the corresponding label. For continuous labels we measure the Mean Squared Error (MSE) and for categorical variables we use Accuracy. The underlined entries indicate the component that should perform best in predicting each label, and the $\Delta$ entries show the performance gap between the representation components. The larger this gap, the better the decoupling. All benchmarks except for M1 label with APOLLO learn the most relevant information for each label in the right representation component. However, in many instances, especially with Simulation II, we see a low $\Delta$ value, showing that information is replicated in the other components as well. MultiLoReFT produces a larger performance gap between correct and incorrect components compared to baselines for the majority of labels, indicating that information has been more cleanly separated and removed from the wrong subspaces. In some cases, like the M2 label for Simulation I, APOLLO achieves a larger gap, but this is mainly due to a generally lower performance for all components. Figure 2 visualizes the subspaces learned by MultiLoReFT, demonstrating that the shared label is clearly separable in the shared space, while each modality-specific label is best separated in its own subspace.

The real data further validate the performance of MultiLoReFT (Table 2) and demonstrate its utility on complex data. In Flickr (M1:Image, M2:Caption), where the label indicates whether captions are in English or French, the modality-specific representation of text is the only component that reliably encodes this information. This is also visible from the well-separated clusters of Caption Language in Figure 3. MultiLoReFT not only captures this information in the right subspace, but also has the largest $\Delta$ among all tested methods. The shared representation performance is random, meaning the language information is completely removed from this subspace. In Crema-D (M1:Video, M2:Audio), which pairs video and audio, we consider three labels: (i) Sentence ID, best captured by the audio modality; (ii) Ethnicity; and (iii) Sex, both conveyed primarily through video. MultiLoReFT aligns each label with the correct component, but also achieves larger performance gaps

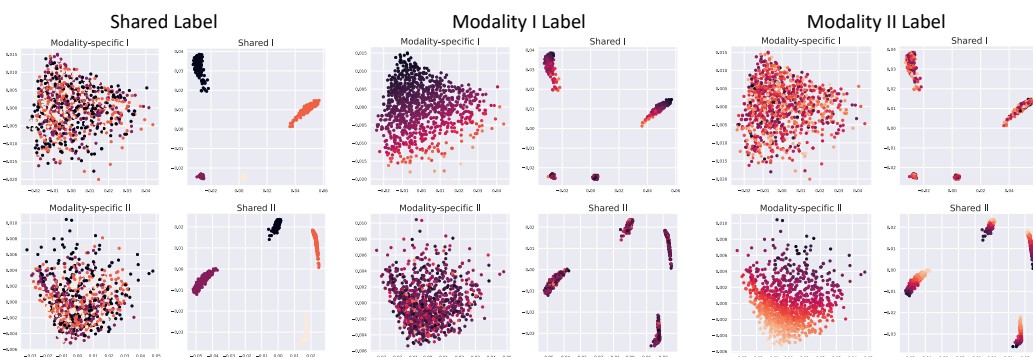

Figure 2: Visualization of subspaces learned by MultiLoReFT on Simulation I. Each panel shows a 2D PCA projection of the shared or modality-specific representations, colored by the underlying generative label. For the shared label (left block), clear clustering emerges in the shared subspace while modality-specific subspaces remain unstructured, indicating successful disentanglement. For the modality I and II labels (middle and right blocks), the corresponding modality-specific subspace captures the structure, whereas the shared subspaces remain agnostic.

between relevant and irrelevant components for Sentence ID and Ethnicity labels. Figure 3 scatter plots also demonstrate this separation.

Table 1: Measuring the decoupling of shared and modality-specific information on simulated data. We report the predictability of different representation components ($\mathbf{z}_s$, $\mathbf{z}_{m1}$, $\mathbf{z}_{m2}$) for the underlying generative variables, using accuracy (Acc) for categorical and mean squared error (MSE) for continuous variables. Large performance gap $\Delta$ indicates better decoupling.

| | | | Simulation I | | | Simulation II | |
|---|---|---|---|---|---|---|
| Model | Rep. | Shared (Acc)↑ | M1 (MSE)↓ | M2 (MSE)↓ | Shared (Acc)↑ | M1 (Acc)↑ |
| MultiLoReFT | $\mathbf{z}_s$ | 1.000±0.000 | 0.048±0.014 | 0.059±0.010 | 1.000±0.000 | 0.537±0.075 |
| | $\mathbf{z}_{m1}$ | 0.798±0.144 | 0.009±0.003 | 0.081±0.006 | 0.500±0.000 | 1.000±0.000 |
| | $\mathbf{z}_{m2}$ | 0.756±0.115 | 0.075±0.007 | 0.005±0.002 | 0.598±0.196 | 0.500±0.000 |
| | $\Delta$ | **0.223±0.130** | 0.038±0.014 | 0.054±0.010 | **0.451±0.098** | **0.463±0.075** |
| DRIM-U | $\mathbf{z}_s$ | 1.000±0.000 | 0.063±0.008 | 0.061±0.007 | 1.000±0.000 | 0.757±0.008 |
| | $\mathbf{z}_{m1}$ | 0.997±0.004 | 0.001±0.000 | 0.083±0.003 | 1.000±0.000 | 1.000±0.000 |
| | $\mathbf{z}_{m2}$ | 1.000±0.000 | 0.081±0.009 | 0.007±0.001 | 0.967±0.006 | 0.500±0.000 |
| | $\Delta$ | 0.001±0.002 | **0.062±0.008** | 0.054±0.007 | 0.033±0.006 | 0.243±0.008 |
| APOLLO | $\mathbf{z}_s$ | 1.000±0.000 | 0.027±0.004 | 0.076±0.009 | 1.000±0.000 | 1.000±0.000 |
| | $\mathbf{z}_{m1}$ | 0.998±0.003 | 0.043±0.005 | 0.086±0.007 | 1.000±0.000 | 1.000±0.000 |
| | $\mathbf{z}_{m2}$ | 1.000±0.000 | 0.087±0.018 | 0.019±0.002 | 1.000±0.000 | 0.500±0.000 |
| | $\Delta$ | 0.001±0.001 | 0.016±0.006 | **0.057±0.009** | 0.000±0.000 | 0.000±0.000 |

A key strength of MultiLoReFT lies in its adaptive rank learning. While benchmark methods require fixing the size of shared and modality-specific representations, we set their dimensionalities equal to the final rank learned by MultiLoReFT. This provides the benchmarks with an advantage to leverage the learned ranks. Because as shown in Table 4 in the Appendix, benchmark performance degrades substantially when dimensionalities are varied, as they tend to redundantly encode information across all components. By contrast, MultiLoReFT automatically prunes the dimensionality of each subspace during training, directly discovering the appropriate structure from the data. Table 7 in the Appendix shows the inital and converged ranks for all experiments and demonstrates consistency across random seeds, yielding compact, stable representations.

## 5.2 LEARNING CROSS-MODAL INFORMATION

Multimodal training enables representations to capture information from complementary modalities, thereby improving downstream predictive performance. We demonstrate that fine-tuning with

Table 2: Measuring the decoupling of shared and modality-specific information on Crema-D and Flickr30. We report classification accuracy for each representation component ($\mathbf{z}_{s1}, \mathbf{z}_{s2}, \mathbf{z}_{m1}, \mathbf{z}_{m2}$) across Sentence identity, Ethnicity, Sex, and Language tasks. High performance concentrated in the corresponding component, with others near chance, reflects effective separation of shared and modality-specific information.

| Model | Rep. | CremaD | | | Flickr30 |
|---|---|---|---|---|---|
| | | Sentence ID (Acc.)↑ | Ethnicity (Acc.) ↑ | Sex (Acc.) ↑ | Language (Acc.)↑ |
| MultiLoReFT | $\mathbf{z}_s$ | 0.303±0.041 | 0.500±0.000 | 0.788±0.033 | 0.500±0.000 |
| | $\mathbf{z}_{m1}$ | 0.244±0.021 | 0.981±0.031 | 1.000±0.000 | 0.500±0.000 |
| | $\mathbf{z}_{m2}$ | 0.975±0.010 | 0.500±0.000 | 0.500±0.000 | 1.000±0.001 |
| | $\Delta$ | **0.672±0.042** | **0.481±0.031** | 0.212±0.033 | **0.500±0.001** |
| DRIM-U | $\mathbf{z}_s$ | 0.540±0.032 | 0.500±0.000 | 0.500±0.000 | 0.699±0.071 |
| | $\mathbf{z}_{m1}$ | 0.201±0.017 | 0.968±0.006 | 0.992±0.003 | 0.500±0.000 |
| | $\mathbf{z}_{m2}$ | 0.980±0.012 | 0.500±0.000 | 0.535±0.046 | 1.000±0.000 |
| | $\Delta$ | 0.440±0.034 | 0.468±0.006 | **0.492±0.003** | 0.301±0.071 |
| APOLLO | $\mathbf{z}_s$ | 0.717±0.014 | 0.500±0.000 | 0.911±0.021 | 1.000±0.000 |
| | $\mathbf{z}_{m1}$ | 0.181±0.012 | 0.954±0.006 | 0.992±0.007 | 0.500±0.000 |
| | $\mathbf{z}_{m2}$ | 0.738±0.018 | 0.500±0.000 | 0.500±0.000 | 1.000±0.000 |
| | $\Delta$ | 0.021±0.023 | 0.454±0.006 | 0.008±0.022 | 0.000±0.000 |

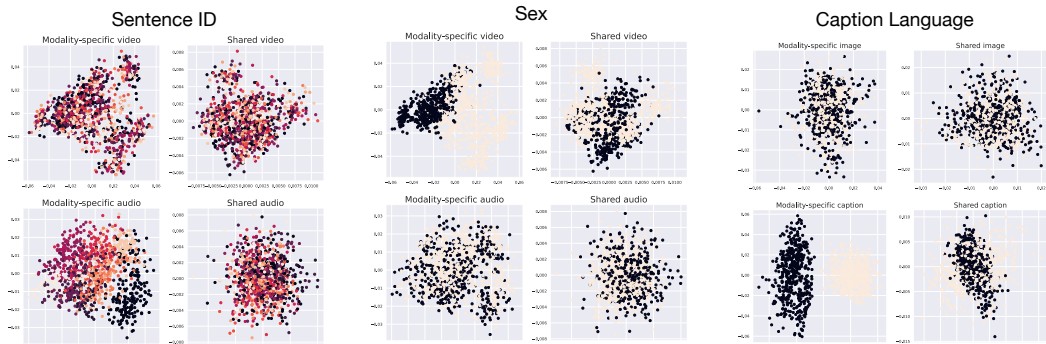

Figure 3: PCA visualizations of learned representations on Crema-D and Flickr30. Task-relevant information (Sentence ID, Sex, Caption Language) is concentrated in the corresponding modality-specific components, while other components remain closer to random structure, illustrating effective disentanglement.

MultiLoReFT yields representations that, when fused, outperform a range of benchmarks (Table 3a). These include both disentanglement-based methods and classical fusion strategies, all applied on top of pretrained embeddings. We also examine the impact of stronger pretrained unimodal encoders on multimodal performance in Appendix the A.5. As expected, more powerful encoders capture richer structure in the data and consequently yield higher-quality multimodal representations.

Beyond fused representations, we show that fine-tuning also improves each individual modality by allowing it to incorporate information from the other. Figure 3b illustrates this effect by comparing the predictive performance of the fine-tuned embeddings $\Phi_1$ and $\Phi_2$ against their pretrained counterparts $h_1$ and $h_2$, as well as against contrastive fine-tuning baselines. Across both modalities, fine-tuning consistently enhances predictive accuracy, with MultiLoReFT achieving the largest gains. Notably, the weaker modality in each pair benefits the most, reflecting its ability to leverage cross-modal information provided by the stronger modality.

|  | (a) | |
|---|---|---|
| Method | Simulation I joint label | Crema-D emotion |
| MultiLoReFT | **0.009±0.002** | **0.460±0.031** |
| APOLLO | **0.009±0.001** | 0.313±0.012 |
| DRIM-U | 0.012±0.001 | 0.443±0.018 |
| Late fusion | 0.054±0.005 | 0.452±0.039 |
| Cross attention | **0.009±0.003** | 0.304±0.032 |
| Contrastive | 0.056±0.006 | 0.419±0.025 |
| MI | 0.016±0.002 | 0.296±0.022 |

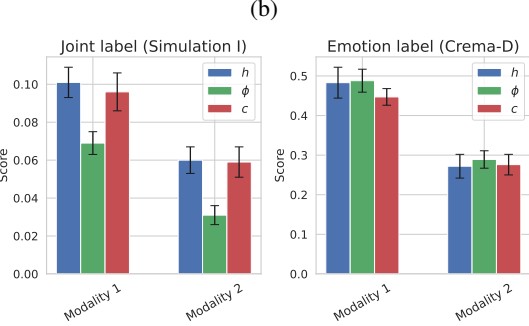

Table 3: Comparison of fusion methods for multimodal prediction on Simulation I (joint label prediction) and Crema-D (emotion prediction). (a) Table reports mean ± std of each baseline predicting the label using the fused representations. (b) Figure shows predictive performance of all modality representations, comparing pretrained representations $h$ with representations fine-tuned with contrastive learning $c$ and using MultiLoReFT $\Phi$, showing how finetuning enhances each representations' predictive power.

## 6 DISCUSSION

This work introduces MULTILOREFT, a low-rank representation fine-tuning framework for multimodal representation learning that disentangles shared and modality-specific information for improved interpretability. Our approach is model-agnostic, only requires learning a small number of parameters, and enables efficient multimodal fusion without sacrificing interpretability. MULTILOREFT uses an adaptive rank pruning that allows the model to learn the proper size of each subspace, improving both performance and insight into information. The broader importance of this contribution lies in its ability to leverage pretrained unimodal encoders for multimodal downstream tasks. By decoupling shared from modality-specific information, MultiLoReFT provides interpretable insights into what each modality contributes, while adaptive rank learning offers a practical solution in settings where the dimensionality of shared and private subspaces is unknown. These properties make the method particularly valuable for scientific discovery domains such as healthcare, where interpretability and data efficiency are critical.

In this work we focus on 2 modalities. While in theory, MultiLoReFT can be extended to multiple modalities by defining a set of projection subspaces for each combination of shared information and expanding the independence constraints across all relevant component pairs, this added granularity could also reduce interpretability as it becomes unclear what constitutes "shared information between two modalities but not a third", or how such partial factors would be utilized in downstream tasks. We therefore focus on the bimodal case for consistency with prior work and lack of ground truth labels for validation, but future work can investigate such extension. Also, MultiLoReFT builds on top of frozen pretrained embeddings, its performance is inherently constrained by the quality and coverage of those unimodal encoders. If the pretrained representations fail to capture modality-relevant information, the gains from fine-tuning will be limited. Future work could explore coupling MultiLoReFT with joint pretraining, scaling to larger multimodal corpora, and extending the framework to more than two modalities.

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

## A  APPENDIX

### A.1  SIMULATED DATASETS

To systematically evaluate the functionality of our approach, we constructed two simulated datasets in which the underlying generative factors are explicitly controlled and well understood. These datasets enable us to verify whether each component of MultiLoReFT captures the intended signal. We provide a brief description of each below.

#### A.1.1  SIMULATION I

Simulation I generates two modalities from a combination of shared and modality-specific latent variables, sampled from non-Gaussian distributions. In total, $n_{\text{hidden}} = 2 + 2 + 2 = 6$ hidden variables are defined: two shared, two private to modality 1, and two private to modality 2.

**Shared latent factors.** The shared variables $z_s \in \mathbb{R}^2$ are sampled from a *binomial* distribution and slightly perturbed with Gaussian noise to introduce variability:

$$z_s \sim \text{Binomial}(1, 0.5) + \mathcal{N}(0, 0.01I).$$

**Modality-specific latent factors.** Each modality has its own private factors drawn from distinct non-Gaussian distributions:

$$z_{m1} \sim \text{Weibull}(1.5) \times 0.3, \qquad z_{m2} \sim \text{Beta}(3, 2).$$

**Labels.** Each data point is annotated with four labels to probe shared and modality-specific information:

$$y_s \in \{0, 1, 2, 3\}, \quad y_{m1} = \sum_j (z_{m1})_j, \quad y_{m2} = \sum_j (z_{m2})_j, \quad y_{\text{cond}} = \begin{cases} \sum_j (z_{m1})_j & \text{if } y_s = 1, \\ \sum_j (z_{m2})_j & \text{if } y_s = 0, \end{cases}$$

where $y_s$ is a multi-class label (0–3) derived from the unique binomial combinations of $z_s$, $y_{m1}$ and $y_{m2}$ are continuous regression targets based on private modality factors, and $y_{\text{cond}}$ conditionally selects between the two modalities.

**Representations.** Observed features are constructed by concatenating shared and modality-specific variables and projecting them into higher-dimensional spaces via learned linear encoders:

$$h_1 = [z_{m1}, z_s]W_1, \qquad h_2 = [z_{m2}, z_s]W_2,$$

where $W_1, W_2$ are projection matrices sampled from a uniform distribution. The output dimensionality is set to 10 per modality.

This design produces paired representations $(h_1, h_2)$ with explicitly defined non-Gaussian structure and labels that probe shared, private, and conditional information, enabling controlled evaluation of disentanglement and multimodal fusion.

### A.1.2 SIMULATION II

Simulation II generates two modalities from five latent variables with structured dependencies that introduce both shared and modality-specific components, as well as partial overlap.

**Shared latent factors.** Two variables define the shared information:

$$z_1 \sim \text{Ber}(0.5), \qquad z_2 = z_1 + \sqrt{0.0045}\,\Gamma(5, 1),$$

where $z_1$ is a binary Bernoulli variable and $z_2$ is a continuous variable correlated with $z_1$ through an additive Gamma perturbation.

**Modality-specific latent factors.** Modality 1 has two private factors:

$$z_3 \sim \text{Ber}(0.5), \qquad z_4 = 2z_2 + z_3 + \sqrt{0.00125}\,\Gamma(2, 2),$$

while Modality 2 has one private factor:

$$z_5 = z_2 + \sqrt{0.0075}\,\Gamma(3, 2).$$

This setup ensures overlap, as $z_2$ influences both $z_4$ and $z_5$, creating cross-modal dependencies while maintaining modality-specific variation.

**Labels.** Labels are directly tied to the latent variables, enabling controlled evaluation of shared versus modality-specific representations. Binary classification tasks can be derived from $z_1$ (shared) or $z_3$ (modality 1 specific), while regression targets can be defined from $z_2$ (shared), $z_4$ (modality 1 specific), or $z_5$ (modality 2 specific).

**Representations.** As in Simulation I, observed features for each modality are constructed by concatenating the corresponding shared and private variables and projecting them into higher-dimensional feature spaces via random linear transformations:

$$h_1 = [z_{m1}, z_s]W_1, \qquad h_2 = [z_{m2}, z_s]W_2,$$

where $W_1, W_2$ are sampled from uniform distributions. The output dimensionality is set to 40 and 80 for each modality.

This design complements Simulation I by introducing structured overlap and dependence between modalities, testing whether models can disentangle shared information from modality-specific signals in the presence of cross-modal dependencies.

## A.2 SUPPLEMENTARY RESULTS

### A.2.1 THE EFFECT OF LEARNING THE RIGHT RANK IN THE PERFORMANCE OF BENCHMARKS

As discussed in the results section, we size benchmark representations to match the ranks learned by MultiLoReFT. This choice is critical, since the appropriate subspace dimensionality is rarely known in advance and strongly affects performance. Table 4 reports benchmark results on simulated settings when their representation size is instead fixed to the initial input dimensionality used by MultiLoReFT. In this case, performance degrades noticeably, with substantial leakage of information across components, underscoring the importance of adaptive rank selection.

Table 4: Benchmark performance on simulated datasets when representation size is fixed to the initial input dimensionality used by MultiLoReFT. Performance degrades compared to using the learned ranks, with clear leakage of information across components, highlighting the importance of learning rank.

| Model | Rep. | Simulation I | | | | Simulation II | |
|-------|------|--------------|--------------|--------------|--------------|---------------|---------------|
| | | Shared (Acc.) | Shared (Sill.) | M1 (MSE) | M2 (MSE) | Shared (Acc.) | M1 (Acc.) |
| APOLLO | $\mathbf{z}_{s1}$ | 1.00±0.00 | 0.838 ± 0.006 | 0.006 ± 0.001 | 0.083 ± 0.012 | 1.000±0.000 | 1.000±0.000 |
| | $\mathbf{z}_{s2}$ | 1.00±0.00 | 0.801 ± 0.007 | 0.080 ± 0.005 | 0.001 ± 0.000 | 1.000±0.000 | 0.500±0.000 |
| | $\mathbf{z}_{m1}$ | 1.00±0.00 | 0.446 ± 0.021 | 0.005 ± 0.002 | 0.089 ± 0.004 | 1.000±0.000 | 1.000±0.000 |
| | $\mathbf{z}_{m2}$ | 1.00±0.00 | 0.487 ± 0.049 | 0.081 ± 0.012 | 0.001 ± 0.000 | 1.000±0.000 | 0.500±0.000 |
| DRIM-U | $\mathbf{z}_{s1}$ | 1.00±0.00 | 0.753±0.005 | 0.037±0.004 | 0.089±0.012 | 0.686±0.021 | 0.500±0.000 |
| | $\mathbf{z}_{s2}$ | 1.00±0.00 | 0.760±0.006 | 0.080±0.013 | 0.032±0.006 | 0.706±0.018 | 0.500±0.000 |
| | $\mathbf{z}_{m1}$ | 1.00±0.00 | 0.293±0.015 | 0.000±0.000 | 0.084±0.006 | 1.000±0.000 | 1.000±0.000 |
| | $\mathbf{z}_{m2}$ | 1.00±0.00 | 0.381±0.007 | 0.082±0.010 | 0.002±0.000 | 1.000±0.000 | 0.500±0.000 |

### A.2.2 ABLATION STUDY FOR PRUNING AND STAGING

To assess the contribution of individual design choices in MultiLoReFT, we conduct ablation experiments removing either the pruning mechanism or the staged training procedure. Results are summarized in Table 5. Both components are critical: eliminating pruning leads to inflated subspace sizes and information leakage across components, while skipping staged training reduces stability and weakens decoupling. The full model, which combines both pruning and staged optimization, consistently achieves the highest predictive accuracy across simulated and real datasets, underscoring their complementary benefits.

Table 5: Ablation study of MultiLoReFT on simulated and real datasets. Removing either the pruning mechanism or the staged training procedure reduces performance, while the full model combining both achieves the best results.

| Model | Rep. | Simulation I | | | Simulation II | |
|-------|------|--------------|--------------|--------------|---------------|---------------|
| | | Shared (Acc.) | M1 (MSE) | M2 (MSE) | Shared (Acc.) | M1 (Acc.) |
| MultiLoReFT | $\mathbf{z}_s$ | 1.000±0.000 | 0.048±0.014 | 0.059±0.010 | 1.000±0.000 | 0.537±0.075 |
| | $\mathbf{z}_{m1}$ | 0.798±0.144 | 0.009±0.003 | 0.081±0.006 | 0.500±0.000 | 1.000±0.000 |
| | $\mathbf{z}_{m2}$ | 0.756±0.115 | 0.075±0.007 | 0.005±0.002 | 0.598±0.196 | 0.500±0.000 |
| MultiLoReFT | $\mathbf{z}_s$ | 1.000±0.000 | 0.002±0.001 | 0.018±0.007 | 1.000±0.000 | 0.500±0.000 |
| (No pruning) | $\mathbf{z}_{m1}$ | 0.994±0.006 | 0.000±0.000 | 0.083±0.007 | 0.500±0.000 | 1.000±0.000 |
| | $\mathbf{z}_{m2}$ | 0.961±0.034 | 0.078±0.008 | 0.004±0.00 | 0.664±0.099 | 0.500±0.000 |
| MultiLoReFT | $\mathbf{z}_s$ | 1.000±0.000 | 0.066±0.013 | 0.075±0.008 | 1.000±0.000 | 0.705±0.165 |
| (No staging) | $\mathbf{z}_{m1}$ | 0.817±0.138 | 0.008±0.003 | 0.082±0.007 | 0.500±0.000 | 1.000±0.000 |
| | $\mathbf{z}_{m2}$ | 0.642±0.269 | 0.076±0.005 | 0.016±0.016 | 0.774±0.229 | 0.500±0.000 |

Furthermore, Table 6 shows the effect of different components of the loss term in the overall performance of MultiLoReFT. Each row presents the results with one component removed. We see that for modality-specific labels, the orthogonality loss alone is sufficient to encourage decoupling, since these targets depend mainly on unimodal geometry—each modality's signal lies on its own manifold, so ensuring linear disjointness prevents interference without needing additional statistical constraints.

In contrast, the shared categorical label relies on both orthogonality and independence, because it emerges from the joint statistical structure across modalities; orthogonality separates the spaces geometrically, while independence removes nonlinear correlations and redundancy, allowing the shared subspace to capture only the truly cross-modal information rather than correlated modality-specific noise. Mutual-information (MI) retention loss preserves unimodal content. Without MI, linear-probe and few-shot performance drop across all subspaces (shared and private).

Table 6: Ablation of the MultiLoReFT objective. The first row is the full model. The next three rows remove the indicated loss term (one at a time). The final row trains with fixed loss weights instead of automatic weighting via GradNorm.

| | | | Simulation I | | | Simulation II | |
|---|---|---|---|---|---|---|---|
| Model | Rep. | Shared (Acc)↑ | M1 (MSE)↓ | M2 (MSE)↓ | Shared (Acc)↑ | M1 (Acc)↑ |
| MultiLoReFT | $\mathbf{z}_s$ | 1.000±0.000 | 0.048±0.014 | 0.059±0.010 | 1.000±0.000 | 0.537±0.075 |
| | $\mathbf{z}_{m1}$ | 0.798±0.144 | 0.009±0.003 | 0.081±0.006 | 0.500±0.000 | 1.000±0.000 |
| | $\mathbf{z}_{m2}$ | 0.756±0.115 | 0.075±0.007 | 0.005±0.002 | 0.598±0.196 | 0.500±0.000 |
| | Δ | **0.223±0.130** | 0.038±0.014 | 0.054±0.010 | **0.451±0.098** | 0.463±0.075 |
| MultiLoReFT | $\mathbf{z}_s$ | 1.000±0.000 | 0.029±0.034 | 0.021±0.009 | 1.000±0.000 | 0.500±0.000 |
| - orthogonality | $\mathbf{z}_{m1}$ | 1.000±0.000 | 0.003±0.000 | 0.081±0.006 | 0.748±0.204 | 1.000±0.000 |
| | $\mathbf{z}_{m2}$ | 0.998±0.004 | 0.072±0.007 | 0.005±0.002 | 1.000±0.000 | 0.500±0.000 |
| | Δ | 0.001±0.002 | 0.026±0.034 | 0.016±0.009 | 0.126±0.102 | 0.500±0.000 |
| MultiLoReFT | $\mathbf{z}_s$ | 1.000±0.000 | 0.064±0.006 | 0.069±0.012 | 1.000±0.000 | 0.599±0.140 |
| - independence | $\mathbf{z}_{m1}$ | 0.999±0.002 | 0.006±0.001 | 0.080±0.006 | 0.735±0.051 | 1.000±0.000 |
| | $\mathbf{z}_{m2}$ | 0.993±0.007 | 0.073±0.007 | 0.005±0.003 | 1.000±0.001 | 0.500±0.000 |
| | Δ | 0.004±0.005 | 0.058±0.006 | **0.064±0.012** | 0.133±0.026 | 0.401±0.140 |
| MultiLoReFT | $\mathbf{z}_s$ | 0.931±0.098 | 0.062±0.018 | 0.063±0.013 | 0.987±0.019 | 0.629±0.095 |
| - MI | $\mathbf{z}_{m1}$ | 0.835±0.066 | 0.006±0.001 | 0.081±0.007 | 0.599±0.071 | 0.746±0.180 |
| | $\mathbf{z}_{m2}$ | 0.850±0.139 | 0.073±0.008 | 0.008±0.004 | 0.570±0.100 | 0.500±0.000 |
| | Δ | 0.089±0.103 | 0.056±0.018 | 0.055±0.013 | 0.403±0.064 | 0.117±0.204 |
| MultiLoReFT | $\mathbf{z}_s$ | 1.000±0.000 | 0.067±0.013 | 0.063±0.021 | 1.000±0.000 | 0.597±0.069 |
| - GradNorm | $\mathbf{z}_{m1}$ | 1.000±0.000 | 0.005±0.001 | 0.079±0.005 | 0.997±0.004 | 1.000±0.000 |
| | $\mathbf{z}_{m2}$ | 0.983±0.025 | 0.079±0.009 | 0.020±0.011 | 1.000±0.000 | 0.500±0.000 |
| | Δ | 0.009±0.013 | **0.062±0.013** | 0.043±0.021 | 0.002±0.002 | 0.403±0.069 |

## A.3 RANK-ADAPTATION

Table 7 reports the initial and converged dimensionalities of the shared ($\mathbf{z}_s$) and modality-specific ($\mathbf{z}_{m1}, \mathbf{z}_{m2}$) subspaces across different datasets. These results are averaged over multiple random seeds, with standard deviations shown to reflect variability. We observe that MULTILOREFT consistently prunes high-dimensional initializations down to compact and stable subspaces, with only minor variation across runs. This consistency highlights that the model is able to reliably identify the rank of shared versus modality-specific information.

The results presented in the main paper are achieved with these compact representations, demonstrating that strong disentanglement and predictive performance do not require large subspace sizes. Instead, the rank adaptation procedure ensures both efficiency and interpretability, by automatically converging to low-dimensional but informative representations across datasets.

Table 7: Shared and modality-specific subspace dimensionalities learned by MultiLoReFT. Entries show the initial rank → converged rank mean and standard deviation on 4 different random seeds

| | Simulation I | Simulation II | Crema-D | Flickr |
|---|---|---|---|---|
| $\mathbf{z}_s$ | $10 \to 5.2 \pm 0.38$ | $40 \to 4.0 \pm 0.70$ | $700 \to 29.75 \pm 7.22$ | $700 \to 310 \pm 3.0$ |
| $\mathbf{z}_{m1}$ | $10 \to 4.6 \pm 0.48$ | $40 \to 2.0 \pm 0.00$ | $700 \to 113.5 \pm 4.36$ | $700 \to 142 \pm 5.0$ |
| $\mathbf{z}_{m2}$ | $10 \to 4.6 \pm 0.48$ | $40 \to 2.2 \pm 0.42$ | $700 \to 196.75 \pm 10.68$ | $700 \to 105 \pm 2.0$ |

## A.4 PARAMETER SIZE COMPARISON

A central motivation behind PEFT methods is to achieve *parameter-efficient fine-tuning*. Rather than updating all weights of large pretrained encoders, recent methods introduce lightweight modules whose size scales with the input representation dimension $d$ and the learned subspace size $d^*$. This allows fair comparison across benchmarks in terms of their parameter overhead.

For **MultiLoReFT**, the number of trainable parameters is on the order of:

$$\#\text{params} \approx 770 \, d \, d^*,$$

corresponding to the projection matrices and small transformation functions. Importantly, we begin with a large parameter space but prune down subspaces dynamically during training, which further reduces the effective size.

For **APOLLO** (Zhang et al., 2024b), the parameter cost includes adaptor layers and explicit sample-wise representations, yielding:

$$\#\text{params} \approx 2048\, d\, d^* \;+\; 3d^* n_{\text{train}},$$

where the second term scales linearly with the number of training samples $n_{\text{train}}$, making the method less efficient for large datasets.

For **DRIM-U** (Robinet et al., 2024), adaptor heads, decoders, and discriminators introduce higher overhead:

$$\#\text{params} \approx 1536\, d\, d^* \;+\; 65.5 d^*.$$

These expressions approximate $d^*$ as the average subspace size across shared and modality-specific components. While exact sizes may vary, the relative scaling highlights the efficiency of Multi-LoReFT, enabling scalable fine-tuning in multimodal settings and making training feasible under limited computational budgets.

### A.5 EFFECT OF UNIMODAL ENCODER STRENGTH ON MULTIMODAL PERFORMANCE

Table 8 compares multimodal performance on different pre-trained encoders on the CREMA-D emotion recognition task. In the first configuration **(Encoders I)**, we use simpler unimodal encoders, Wav2Vec 2.0 Base pretrained on Librispeech-960h (Baevski et al., 2020) for audio and 3D ResNet-18 pretrained on Kinetics-400 (He et al., 2016; Kay et al., 2017) for video. In the second configuration **(Encoders II)**, we replace these with stronger encoders, MViT-V2-S pretrained on Kinetics-400 (Li et al., 2022) for video and WavLM-Base+ for audio (Chen et al., 2022).

As shown, all multimodal baselines improve with higher-capacity unimodal encoders, but the gains are more pronounced for MultiLoReFT. Importantly, the relative ordering of methods remains stable, indicating that MultiLoReFT's advantage is complementary to the underlying encoder strength. This property is an advantage of methods that leverage unimodal encoders, as unimodal models continue to advance, their improved representational quality can be benefited from to construct higher-performing multimodal representations, even under limited multimodal supervision.

Table 8: The effect of Unimodal Encoder Strength on Multimodal Performance on Crema-D dataset for emotion detection. Encoders I setup uses relatively simpler video and audio encoders while Encoders II setup uses more advanced pretrained models.

| Method | Crema-D emotion Encoders I | Crema-D emotion Encoders II |
|---|---|---|
| MultiLoReFT | **0.460±0.031** | **0.810±0.017** |
| APOLLO | 0.313±0.012 | 0.534±0.017 |
| DRIM-U | 0.443±0.018 | 0.733±0.008 |
| Late fusion | 0.452±0.039 | 0.792±0.017 |
| Cross attention | 0.304±0.032 | 0.795±0.022 |
| Contrastive | 0.419±0.025 | 0.764±0.009 |
| MI | 0.296±0.022 | 0.791±0.022 |

