# OpenReview forum: "Decoupling Shared and Modality-Specific Subspaces in Multimodal Learning via Low-Rank Representation Fine-Tuning"
_ICLR.cc/2026/Conference — Submitted to ICLR 2026_

### Official Review · Reviewer_JLeH · 2025-10-25

**Soundness:** 2
**Presentation:** 2
**Contribution:** 3
**Rating:** 4
**Confidence:** 4

**Summary:**

The paper proposes MultiLoReFT, an efficient low-rank finetuning framework that disentangles shared and modality-specific information for multimodal representation learning. This offers more interpretability and control over the representation learning. The proposed approach is compared with two other methods APOLLO and DRIM-U and has shown to be more effective in decoupling the representation on both controlled, synthetic tasks and real-world tasks CremaD and Flickr30.

**Strengths:**

- The paper is well-motivated by the research question of decoupling shared and modality-specific information in multimodal representation learning and offers an efficient approach that gives more interpretability and control over the representation learned;
- The method, synthetic data generation, and experiment setups are well-documented;
- The design of components of loss that concern with the independence, orthogonality, and mutual information is novel;
- Ablation of the proposed method on controlled, synthetic tasks and real-world tasks with different fusion schemes, along with the PCA visualization of the learned representations, show the effectiveness of the proposed method in decoupling the shared and modality-specific representations;

**Weaknesses:**

- The paper mentions 2 key questions at the beginning of Section 4 Experiments that the reviewer agrees are very important to be addressed: 1. MultiLoReFT effectively captures the shared and modality-specific information and gives an interpretable representation showing how these information are distributed across the components; 2. The decoupled representation from MultiLoReFT can effectively improve multimodal representations to better perform at downstream tasks. While Section 5 Results gives an in-depth study to answer 1, question 2 is not addressed. In particular, evaluations of the decoupled representations used jointly in downstream tasks (with proper ablations on model architecture, scales, tasks preferred) are missing, which has significantly undermined the utility and motivation of the work (note this is different from table 2, which evaluates the effectiveness of individual components with knowing which component works the best a priori and can only indicate the effectiveness of the framework in decoupling representation);
- One highlight / contribution of MultiLoReFT is its parameter-efficiency. While it's clear from the context that this approach does not require training the encoders, there lacks an explicit section discussing / comparing the compute between MultiLoReFT and other existing approaches;
- The writing can be further improved: in particular,
1. The results section is a little unorganized in terms of its presentation. Preferably, there should be a one-sentence summary highlighting the main takeaway from the discussed results;
2. $\textbf{z}_{m_i}$ are not labeled clearly for table 2 to indicate which captures video, audio, or text, image in different tasks;
3. It's unclear what metrics are evaluated in table 3 and why the lower scores indicate stronger predicative power;
4. Inconsistent naming of the method name: some are in small caps MultiLoReFT while others are the regular text MultiLoReFT;

**Questions:**

- The authors should work on including more evaluations of the learned, decoupled representations in downstream tasks: would the more separated representations allow the model to better perform at various multimodal predictions? It's understandable that this is not the focus of the paper, but the paper should include a minimal ablation on the effect of model architecture, scale, modalities/tasks on the effectiveness of MultiLoReFT. Otherwise, it is not clear why it's important to learn a more effectively decoupled representation for multimodal learning except for better interpretability;
- Why is Simulation II only has the M1 prediction task and it's evaluated in Accuracy as opposed to MSE in Simulation I M1? What's the difference? Similarly, why is only the Language task evaluated for Flickr30 and is there a task that can be dominantly predicted using $\textbf{z}_{m_2}$?

---

> ### Author Response · Authors · 2025-11-19
> **Rebuttal response**
>
> We thank the reviewer for their thoughtful and constructive feedback, and we have addressed all concerns and questions point-by-point in the responses below.
>
> 1. W1: We appreciate emphasizing these two key questions. We agree that both are important, and we clarify that the current work is primarily focused on addressing Question 1, while also laying the groundwork for Question 2, which concerns downstream integration and performance.
> That said, we recognize the importance of understanding how the joint decoupled representation performs in downstream multimodal tasks. To partially address this, Table 3a evaluates the fused finetuned representation on emotion recognition and the joint label in Simulation I, showing that combining the disentangled components yields competitive multimodal performance. Moreover, in new experiments included in the Appendix, we replicated the CREMA-D setup using stronger unimodal encoders, MViT-V2-S (K400) for video and WavLM-Base+ for audio, and observed clear downstream gains, approaching leaderboard performance. These results demonstrate that MultiLoReFT can scale naturally with encoder quality, improving downstream accuracy without sacrificing interpretability or requiring additional fine-tuning of encoders.
> We will make this relationship between interpretability and downstream potential more explicit in the revised manuscript and highlight the new results showing that stronger unimodal backbones translate directly into improved multimodal task performance.
>
> 2. W2: Thank you for raising this because parameter/compute efficiency is indeed a key contribution of MultiLoReFT. Our method freezes all unimodal encoders and trains only lightweight projection heads and adapters, which yields substantially lower trainable parameter count relative to approaches that fine-tune encoders or add deep fusion stacks. We already report trainable-parameter counts for all baselines in the appendix, showing that MultiLoReFT has orders of  magnitude fewer parameters compared to other decoupling baselines. In the revised manuscript we will  make this reference explicit in the main text.
>
> 3. W3: Thank you for pointing these writing notes out. We have added a summary of results at the beginning of the evaluation section to give an overview of the evaluation, added clear labels to modalities reported in the tables, as we have described the metrics in the text in addition to the captions.
>
> 4. Q1: MultiLoReFT promotes interpretability in multimodal representations by explicitly disentangling shared and private subspaces. The primary value is not merely to show that different components may boost specific downstream tasks, but to enable knowledge discovery: it reveals what each modality uniquely contributes and what is redundantly shared. In our datasets, for example, probing each component shows that language is encoded in the caption and is non-identifiable from images. We selected such intentionally simple cases that would allow us to validate the framework’s functionality. In real-world settings, such disentanglement provides actionable insight; for instance, in healthcare, it can identify which clinical modality (e.g., imaging, physiological signals) carries unique diagnostic value, guiding which measurements to acquire to improve predictive power or reduce acquisition cost.
>
> 5. Q2: We appreciate the opportunity to clarify this. In our simulations, M1 and M2 refer to the synthetic labels generated for each setup, each designed to depend primarily on one modality. The exact formulation of these labels varies between simulation settings and their different generative processes, as described in the Appendix. Specifically, in Simulation I, both M1 and M2 are continuous-valued targets (regression tasks), hence they are evaluated using mean squared error (MSE). In contrast, Simulation II uses a categorical M1 label, which makes classification accuracy the appropriate metric.
>
> For Flickr30K, the available annotations contain only language-based (caption) labels, with no corresponding image-only labels that could serve as a distinct modality-specific task. Therefore, we focus on evaluating the language-related component, which allows us to validate whether the model properly encodes textual versus shared multimodal semantics.

---

> ### Comment · Reviewer_JLeH · 2025-11-22
>
> Thanks for the author's responses to the concerns above. Most of the original concerns are properly addressed. There are a few additional comments (which actually belongs to the same concern):
>
> W1: Table 3a and Appendix A.5 do touch on Question 2. Since the beginning of section 4 sets up equal expectation for both questions, the paper should either highlight the results with more discussion about Table 3a and Appendix A.5 (which is not referenced anywhere in the main text), or clarify that the focus is more on Question 1 and only slightly touches 2.
>
> Q1: The reviewer recognizes the value of decoupling shared / modality-specific information for understanding modality importance for practical reasons e.g. cost-efficient data collection. However, if the paper is positioned in this way, MultiLoReFT should be compared with the line of work e.g. [1] which performs the decomposition (even without training). The reviewer believes the value of MultiLoReFT differentiates from e.g. [1] in that it produces shared / modality-specific representations instead of only giving an estimate of the importance. Therefore, the reviewer originally raised the point about showing the utility of the representations by evaluating them on downstream tasks.
>
> So the only concern left is that the reviewer encourages the authors to think about how the paper should be positioned: whether the primary focus is on Question 1, in which case MultiLoReFT should be differentiated from the contribution of e.g. [1,2,3] (this is also mentioned by Reviewer pWRG in their comments), or both Question 1 and 2, in which case the evaluation of the representation on downstream tasks should be much more highlighted. And in either case, the paper should make that explicit.
>
> [1] Liang, Paul Pu, et al. "Quantifying & modeling multimodal interactions: An information decomposition framework." Advances in Neural Information Processing Systems 36 (2023): 27351-27393.
>
> [2] Liang, P. P., Deng, Z., Ma, M. Q., Zou, J. Y., Morency, L. P., & Salakhutdinov, R. (2023). Factorized contrastive learning: Going beyond multi-view redundancy. Advances in Neural Information Processing Systems, 36, 32971-32998.
>
> [3] Dufumier, B., Castillo-Navarro, J., Tuia, D., & Thiran, J. P. (2025). What to align in multimodal contrastive learning?. International Conference on Learning Representations.

---

> > ### Author Response · Authors · 2025-11-24
> >
> > We thank the reviewer for taking the time to read our response, and we are glad that it addressed the main concern. As suggested, we will make the positioning and focus of the paper much clearer by emphasizing that the primary contribution is multimodal disentanglement (Q1), while the downstream results (Q2) serve as a proof of concept rather than the central goal. We will revise the evaluation section accordingly, noting that the first set of experiments targets disentanglement directly, and the second set illustrates how the resulting representations can be used for multimodal learning.
> >
> > We will also update the "Disentangled Multimodal Representations" section of the related work as follows to more accurately situate our contributions within the landscape of existing PID-based and factorization approaches:
> >
> >
> > Disentangled Multimodal Representations: A complementary line of work aims to separate shared and modality-specific information in multimodal settings using information-theoretic tools.[1] quantify redundancy, uniqueness, and synergy between modalities and labels primarily as a diagnostic framework rather than to learn reusable factorized encoders. Other PID-inspired methods such as FactorCL [2] learn task-dependent shared and unique components through end-to-end contrastive objectives. Triple Disentanglement [3] further decomposes representations into shared, relevant, and irrelevant modality-specific components via transformer-based encoder–fusion architectures. CoMM [4]investigates what to align in multimodal contrastive learning, focusing on redundancy and uniqueness within a single shared space, without exposing explicit modality-specific subspaces. In contrast, our work leverages pretrained unimodal encoders to learn explicit shared and private subspaces, with a focus on interpretability and knowledge discovery, rather than optimizing for task-specific performance alone.
> >
> > [1] Liang, Paul Pu, et al. "Quantifying & modeling multimodal interactions: An information decomposition framework." Advances in Neural Information Processing Systems 36 (2023): 27351-27393.
> >
> > [2] Liang, P. P., Deng, Z., Ma, M. Q., Zou, J. Y., Morency, L. P., & Salakhutdinov, R. (2023). Factorized contrastive learning: Going beyond multi-view redundancy. Advances in Neural Information Processing Systems, 36, 32971-32998.
> >
> > [3] Zhou, Y., Liang, X., Chen, H., Zhao, Y., Chen, X., & Yu, L. (2025). Triple disentangled representation learning for multimodal affective analysis. Information Fusion, 114, 102663.
> >
> > [4] Dufumier, B., Castillo-Navarro, J., Tuia, D., & Thiran, J. P. (2025). What to align in multimodal contrastive learning?. International Conference on Learning Representations.

---

> > > ### Comment · Reviewer_JLeH · 2025-11-25
> > >
> > > Thanks for the authors' follow-up. I raised my score to 6.

---

### Official Review · Reviewer_NKz5 · 2025-10-28

**Soundness:** 2
**Presentation:** 3
**Contribution:** 2
**Rating:** 4
**Confidence:** 4

**Summary:**

This paper proposes MultiLoReFT: a low-rank representation fine-tuning framework for multimodal learning based on pre-trained unimodal models. It extends low-rank representation fine-tuning to multimodal scenarios and learns interpretable projection subspaces that separate shared and modality-specific information. MultiLoReFT adaptively learns the rank of each subspace to best capture the complementary contributions of each modality with a minimum number of trainable parameters

**Strengths:**

1. The motivation is clear and targeted, addressing core pain points in multimodal learning
2. The method design is innovative and logically coherent, with clear advantages in efficiency and interpretability
3. The experimental verification is sufficient and rigorous, supporting conclusions effectively

**Weaknesses:**

1. The model used in the experiment doesn't seem very large, so efficient fine-tuning doesn't seem necessary.

2. What about the performance when scaling to larger models like llava?

3. The performance gain of MultiLoReFT in real datasets, such as cramed, seems negligible (0.6%). The MultiLoReFT seems to be useless

4. More importantly, the performance of MultiLoReFT on the CRAMED dataset is even worse than that of a model trained directly with ResNet-18 as the backbone[1]. So what is the point of using such a complex model (DINO) for fine-tuning?


[1] Xiaokang Peng. Balanced Multimodal Learning via On-the-fly Gradient Modulation Xiaokang

**Questions:**

see the weakness

---

> ### Author Response · Authors · 2025-11-19
> **Rebuttal response**
>
> We thank the reviewer for their comments. We would like to clarify one point, that the unimodel encoder models in the paper were chosen primarily as illustrative testbeds to demonstrate the effect of representation-level fine-tuning, rather than to maximize absolute benchmark performance. MultiLoReFT is model-agnostic and its principles apply equally to small or large encoders. Thus, the scale of the backbone (e.g., ResNet-18 or LLaVA) is not critical to the method itself; what matters is the structure of the representation space it learns to refine.
>
> To address the reviewer’s point, we have replicated the CREMA-D experiments using stronger and much larger unimodal encoders: MViT-V2-S pretrained on Kinetics-400 for video [1] and WavLM-Base+ for audio [2]. As expected, the enhanced unimodal backbones yielded higher multimodal performance, approaching state-of-the-art leaderboard results of 0.81 percent accuracy. These results are now included in the appendix (Table 8). This confirms that MultiLoReFT benefits directly from scaling unimodal representations, and that the framework can readily exploit more capable encoders when available.
>
> Importantly, this scalability aligns with the core motivation of MultiLoReFT: to provide a lightweight, unsupervised fine-tuning mechanism that can adapt increasingly powerful pretrained unimodal models to multimodal settings where labeled data are scarce. By aligning and disentangling pretrained embeddings rather than retraining full multimodal architectures, MultiLoReFT offers an efficient and modular way to integrate representation improvements across modalities
>
> [1] Li, Yanghao, et al. "Mvitv2: Improved multiscale vision transformers for classification and detection." Proceedings of the IEEE/CVF conference on computer vision and pattern recognition. 2022.
>
> [2] Chen, Sanyuan, et al. "Wavlm: Large-scale self-supervised pre-training for full stack speech processing." IEEE Journal of Selected Topics in Signal Processing 16.6 (2022): 1505-1518.

---

### Official Review · Reviewer_iJwS · 2025-10-29

**Soundness:** 2
**Presentation:** 2
**Contribution:** 2
**Rating:** 2
**Confidence:** 3

**Summary:**

In this paper, the authors presented MultiLoReFT, a method for low-rank multimodal fine-tuning. The proposed method includes a lightweight architecture that takes in unimodal representation vectors, and transform them into low-rank subspaces while performing nonlinear transformations within the subspace. The authors devised a 3-component training objective that enforces independence between shared/modality-specific representations, the mutual information, and orthogonality between low-rank transformations. In addition, pruning was applied to rank adaptation. The proposed method was evaluated against 2 baseline finetuning methods (DRIM-U and APOLLO) on 2 simulated datasets as well as 2 real datasets, and the evaluations focused on assessing the disentangling of shared and modality-specific signals. Ablation studies was conducted on pruning and staged training.

**Strengths:**

1. The proposed method is lightweight and have a relatively low number of training parameters.

2. The evaluations show that the proposed method better disentangles shared information and modality-specific information in the generated representations.

3. The inclusion of algorithm blocks on page 5 made the overall methodology a lot more clear and easy to follow.

**Weaknesses:**

1. Limited evaluation on real multimodal datasets: other than the synthetic simulations, the paper only contains experiments on 2 real datasets, out of which only one dataset's main task (CremaD emotion prediction) performance was compared to baselines. The main reason of applying multimodal fine-tuning, in most occasions, is to improve model performance over the main task, but the proposed method only showed very marginal gain in CremaD's emotion prediction over a simple late fusion, and no main multimodal classification task was evaluated over Flickr30K.

2. The proposed method also does not seem to consistently show larger deltas in the simulation experiments. The authors claims that APOLLO's larger gap over M2 is due to its overall worse performance, but DRIM-U seem to have same delta on M2 MSE and a much larger delta on M1 MSE for simulation 1.

3. The proposed method seems restricted to 2-modal settings only and does not seem to be generalizable to tasks with more than 2 modalities or tasks with unimodal representations as sequence of vectors rather than single vectors.

4. There is no ablation on some of the most important design choices. For example, there is no ablation on the objectives, so it is unclear whether all 3 objectives are necessary for the proposed method; there is no ablation on doing GradNorm vs fixed objective weighting as hyperparameters. It is also very hard to interpret the results in the existing ablation studies in Table 5, as the results for the full method is not present in the same table.

5. The writing quality is not the best. There are some undefined variables in equation 7: (a) $\tau$ is not officially defined. Do you treat $\tau$ as a fixed constant or a trainable temperature parameter (as in CLIP)? (b) N is not officially defined. Since j is used to indicate modality, maybe N should just be 2? Also, there are typos like "sdsptive" in line 201

**Questions:**

For Table 3a, how exactly are the classification performance obtained? Do you fit another linear layer or MLP on top of the fine-tuned representations ($z_s$)?

In Tables 1 and 2, performance with $z_s$ is reported; however, in Algorithm 1, it only shows how $z_{s1}$ and $z_{s2}$ are obtained. How do you obtain $z_s$ from $z_{s1}$ and $z_{s2}$? Also, Eq(1) states that  $z_{s1}$ and $z_{s2}$ are always equal. Is that true? If so, why is that true?

---

> ### Author Response · Authors · 2025-11-19
> **Rebuttal response**
>
> We thank the reviewer for their thoughtful and constructive feedback, and we have addressed all concerns and questions point-by-point in the responses below.
>
> 1. W1: We agree that our primary focus differs from most multimodal works that emphasize task performance. The goal of MultiLoReFT is to achieve interpretable multimodal representations, enabling analysis of how information is distributed across shared and modality-specific components. The experiments are therefore mainly structured to validate disentanglement rather than to optimize benchmark scores, hence our focus on datasets with some known underlying factors. Disentangling the shared and private subspaces enables knowledge discovery, it reveals what information each modality uniquely contributes and what information is redundant. In real-world settings this disentanglement can provide actionable insight. For example, in healthcare, it can highlight which clinical modality (e.g., imaging or physiological signals) carries unique diagnostic value, thereby informing which measurements are most critical for improving predictive power or reducing acquisition cost.
>
> 2. W2: The results in Table 2 show that MultiLoReFT achieves consistently strong shared representation accuracy with higher statistical significance while maintaining competitive or superior modality-specific reconstruction accuracy. Learning the proper shared representation is the more critical task, because an easy shortcut for models can be to put all information about one modality into the modality-specific representation, and fail to capture anything in the shared space. This results in a big delta for M1 and M2, since modality-specific representation performance will be high, while the shared will be low because no meaningful information was embedded in it. Therefore it is critical for a model to have a good performance on shared representations, to then justify its superior performance on modality-specific as well.
>
> 3. W3: Evaluating beyond two modalities is an interesting extension. In theory, MultiLoReFT can be extended to multiple modalities by defining a set of projection subspaces for each modality pair and expanding the independence constraints across all relevant component pairs. However, this added granularity could also reduce interpretability as it becomes unclear what constitutes “shared information between two modalities but not a third,” or how such partial factors would be utilized in downstream tasks. We therefore focus on the bimodal case for consistency with prior multimodal disentanglement and PID-based frameworks work, but note that the formulation is conceptually extensible. We have added this to the updated discussion.
>
> 4. W4: Component-wise ablation is be an interesting addition to the paper. We now added loss-component ablations to isolate the roles of orthogonality, independence, and cross-modal MI as well as the weighting. The table is included in the update Appendix.
> In summary, we see that for modality-specific labels, the orthogonality loss alone is sufficient to encourage decoupling, since each modality’s signal lies on its own manifold, so ensuring linear disjointness prevents interference without needing additional statistical constraints.
> In contrast, the shared label relies on both orthogonality and independence, because it emerges from the joint statistical structure across modalities; orthogonality separates the spaces geometrically, while independence removes nonlinear correlations and redundancy, allowing the shared subspace to capture only the truly cross-modal information rather than correlated modality-specific noise. Finally, without MI, linear-probe and few-shot performance drop across all subspaces.
> We also added the main table results to the staging/pruning ablation table in the appendix to make the comparison easier.
>
> 5. W5: We have fixed the typos and clarified the notation in the text.
>
> 6. Q1: For Table 3a, we assess classification performance by training a logistic regression classifier on top of the fused representation, i.e. the concatenation of the fine-tuned representations.
>
> 7. Q2: Thank you for pointing this out, this is an important clarification. The shared representation $z_s$​ reported in Tables 1 and 2 is computed as the mean of the shared embeddings inferred from each modality.
> Equation (1) expresses the characteristics of projections into subspaces that both modalities should map to the same shared representation​. This is the property the model is explicitly optimized to achieve through the alignment and independence losses. In practice, small deviations may remain due to data noise or optimization dynamics, and we therefore take the mean as a stable estimate of the shared embedding used in evaluation. We will make this clearer in the revised version.

---

### Official Review · Reviewer_pWRG · 2025-10-31

**Soundness:** 3
**Presentation:** 2
**Contribution:** 2
**Rating:** 2
**Confidence:** 4

**Summary:**

This paper addresses an important challenge in multimodal machine learning: disentangling the respective contributions of inter- and intra-modal interactions, framed as shared and modality-specific components. The authors propose MultiLoReFT, a low-rank representation fine-tuning framework that facilitates efficient and interpretable multimodal learning by leveraging pretrained unimodal models. The approach extends low-rank adaptation techniques (LoReFT) to the multimodal domain, introducing projection subspaces that separate shared and modality-specific information through independence and orthogonality constraints.

The experimental section includes both controlled simulations and large-scale datasets, and the results suggest that the proposed method achieves a better disentanglement of modality contributions compared to the chosenbaselines. Nonetheless, the paper would benefit from more detailed analyses, such as ablation studies on the contribution of individual loss terms—and additional clarification on certain aspects of the experimental setup.

Overall, the paper presents an interesting and promising direction for interpretable multimodal learning. However, some methodological choices could be better justified, and the experimental validation could be strengthened to fully support the claimed advantages of the approach.

**Strengths:**

1. **Interesting and timely topic of disentanglement.**
The paper addresses an important problem in multimodal machine learning: the disentanglement of shared versus modality-specific representations. This topic is both theoretically significant and practically relevant, as achieving clearer separation between modality interactions can lead to better interpretability, and transferability of multimodal systems.

2. **Well-structured and diverse experimental setting.**
The experimental design is carefully structured to evaluate the proposed framework in both controlled and realistic conditions. The inclusion of simulated datasets allows for a clear and interpretable assessment of the method’s disentanglement capabilities under known ground-truth conditions. Complementing this with experiments on larger-scale, real-world datasets demonstrates the  practical potential of the approach.

3. **Parameter efficiency and flexibility.**
The framework is designed to be computationally efficient by introducing only a limited number of additional parameters through adaptive low-rank subspaces. Moreover, the approach can, in principle, be applied on top of _any_ pretrained unimodal model, which makes it flexible across modalities and backbones. However, while this adaptability is promising, the paper provides limited empirical evidence on how well the framework generalizes across different pretrained representations or scales to substantially larger models.

**Weaknesses:**

1. **Insufficient justification for key constraints and design choices.**
The introduction of independence and orthogonality constraints is intuitively motivated but not sufficiently justified from a theoretical or empirical standpoint. It remains unclear why these specific constraints are optimal for disentangling shared and modality-specific representations, and how sensitive the results are to their exact formulation.

2. **Limited contextualization within related multimodal literature.**
The paper would benefit from a clearer discussion situating the proposed approach within the broader landscape of multimodal representation learning. In particular, several recent methods explicitly aim to model shared and modality-specific information through mutual information (MI)–based formulations, such as FactorCL [1], or through partial information decomposition (PID)–based frameworks, such as CoMM [2]. These families of models also address the problem of disentangling inter- and intra-modal contributions, yet their conceptual relationship with MultiLoReFT is not discussed, and they are not included among the baselines. A more explicit discussion, highlighting similarities, differences, and potential complementarities, along with empirical comparisons, would help clarify the novelty and distinct contribution of MultiLoReFT in relation to existing multimodal approaches.

[1] Liang, P. P., Deng, Z., Ma, M. Q., Zou, J. Y., Morency, L. P., & Salakhutdinov, R. (2023). Factorized contrastive learning: Going beyond multi-view redundancy. Advances in Neural Information Processing Systems, 36, 32971-32998.

[2] Dufumier, B., Castillo-Navarro, J., Tuia, D., & Thiran, J. P. (2025). What to align in multimodal contrastive learning?. International Conference on Learning Representations.


3. **Unclear experimental design and training procedure.**
The description of the training pipeline lacks clarity. It is not explicitly stated whether the model is subsequently fine-tuned for specific downstream tasks after the self-supervised stage, and if so, whether modality fusion occurs during or after fine-tuning. Additionally, the loss configuration across different training stages is not clearly explained — it is uncertain whether the same loss functions are applied throughout, what the “selected convergence criteria” refer to, and whether any ablation was performed on these procedural choices.
Similarly, the role and definition of the cross-modal mutual information loss (l. 183) would benefit from clearer explanation and justification.

4. **Missing ablation studies and analytical evaluations.**
The paper lacks ablation studies that would help disentangle the contribution of each component in the overall loss function. Understanding how each loss term (e.g., independence, orthogonality, mutual information) impacts the final representation quality and disentanglement would significantly clarify the method’s internal dynamics.

5. **Restricted evaluation to two modalities.**
The proposed method is evaluated only on bimodal setups, which limits the generality of the conclusions. Moreover, the formulation does not appear to scale naturally beyond two modalities. Extending the approach to more modalities seeems cumbersome, as the number of pairwise interactions and constraints would likely grow combinatorially. A discussion or preliminary experiment addressing this scalability aspect would strengthen the paper’s claims of general applicability.

6. **Practical limitations in real-world settings.**
In practical multimodal scenarios, the dominant source of information relevant to a given task is typically unknown. The proposed method seems to assume that the relative importance of each modality is known (in all experiments, the dominant feature is underscored), but this assumption may not hold in real applications. It remains unclear how the model would perform when the task-relevant modality is not dominant or when the useful information is distributed unevenly across modalities. Clarifying how the framework handles such cases would help assess its robustness and practical usability.

7. Related to the previous point, Table 3a appears to be the only experiment in which the dominant feature is not known beforehand. However, the description of this experiment lacks clarity. The authors refer to the prediction as being based on “fused” features but do not specify how this fusion is performed or which features are actually used. A more detailed explanation of the fusion mechanism is needed to properly assess the experimental setup and interpret the reported results.

8. **Fairness of comparisons.**
The fairness and consistency of experimental comparisons are not fully transparent. It is not clear whether all baselines were trained under comparable settings, especially regarding the use of pretrained encoders and the amount of supervision. For instance, l.314 mentions that DRIM-U was adapted to a self-supervised variant, without any details on how this affects its performance.

9. **Limited analysis of dependence on encoder quality.**
Since the method relies on pretrained unimodal encoders, it would be interesting to study how performance scales with encoder quality. It remains unclear whether the disentanglement and downstream performance would improve linearly (or saturate) with stronger unimodal representations. This sensitivity analysis is missing.

10. **Potential inconsistencies.**
L. 426 seems to contradict the formulation of the loss function. How can fine-tuning improve individual modality representation if the independence loss was specifically designed to prevent leakage of information. This suggests possible inconsistencies in the implementation or interpretation.

**Questions:**

- Could you elaborate on the motivation for enforcing both independence and orthogonality constraints?

- How does MultiLoReFT conceptually and empirically differ from mutual information–based approaches such as FactorCL [1] or partial information decomposition–based models like CoMM [2]? They could also be included in the comparisons as baselines.

- Can you clarify the overall training procedure? Specifically, after the self-supervised stage, is the model fine-tuned for specific downstream tasks?

- Are the same loss functions used throughout all stages of training, or are they modified across phases?

- Could you provide more details on the definition and role of the cross-modal mutual information loss (l.183)?

- What exactly is meant by the “selected convergence criteria”?

- How would the complexity (in terms of number of constraints or pairwise interactions) scale with the number of modalities? Have you considered any strategies to mitigate potential combinatorial growth?

- How would the framework behave in real-world cases where the dominant or most informative modality is not known a priori?

- Could you clarify how the “fused” features are constructed in Table 3a? What mechanism is used (e.g., concatenation, learned fusion, or averaging)? How does this fusion process affect interpretability and performance?

- How sensitive is the method to the quality of the pretrained unimodal encoders? Does the disentanglement or downstream performance improve consistently with stronger encoders?

- Line 426 seems contradictory to the loss formulation: if the independence loss prevents information leakage, how can fine-tuning improve single-modality representations?

---

> ### Author Response · Authors · 2025-11-19
> **Rebuttal response**
>
> We thank the reviewers for their insightful comments and valuable feedback. Below, we would like to clarify a few misunderstandings and address the main questions raised.
>
> 1. Insufficient justification for key constraints and design choices (also Q1, Q5): We are more than happy to provide the theoretical justification behind our design choices. We have also added this clarification to the updated manuscript.
> The Hilbert--Schmidt Independence Criterion (HSIC) is a nonparametric measure of statistical dependence between two random variables. When the kernels are characteristic (e.g., Gaussian RBF or Laplace), HSIC(X,Y) is 0 if and only if $X$ and $Y$ are statistically independent. Hence, in our method we use an empirical, unbiased estimator of H that we can minimize during training to enforces nonlinear independence between our representation components ($\widehat{\mathrm{HSIC}}(X,Y) = \frac{1}{(n-1)^2} \, \mathrm{tr}(KL)$).
>
> This discourages shared statistical structure across subspaces, but HSIC acts only in the distributional sense and does not guarantee that the learned directions are geometrically distinct. To ensure geometric disjointness, we penalize cross-interactions between shared and private subspaces via the orthogonality loss, by measuring the Frobenius norm of the projection matrices. Minimizing $\mathcal{L}_{orth}$ suppresses these inner products, removing linear overlap (leakage) in subspaces and yielding geometrically disjoint subspaces.
>
> While HSIC guarantees that subspaces do not carry redundant information, orthogonality ensures that their basis vectors do not overlap in representation space. For instance, two statistically independent variables could still align geometrically (colinear bases), and two orthogonal directions could still exhibit nonlinear dependence.
>
> Mutual-information retention preserves the information from the pretrained unimodal embeddings $H$ during disentanglement. We include an InfoNCE-based mutual-information objective. The InfoNCE loss provides a lower bound on the true mutual information ($I(Z;H) \ge \log N - L_{InfoNCE}$ ) where $L_{\text{InfoNCE}} = - \frac{1}{2} \sum_{i=1}^2  \log \frac{\exp(\langle h_i, z^{(i)} \rangle / \tau)} {\sum_{j=1}^N \exp(\langle h_i, z^{(j)} \rangle / \tau)}.$
> Maximizing this bound (i.e., minimizing $\mathcal{L}_{\text{InfoNCE}}$) preserves high mutual information between the projections and their sources. Therefore, the disentangled shared and modality-specific projections remain sufficient summaries of their original embeddings while allowing orthogonality and independence to reshape their structure.
>
>
> 2. Missing ablation studies and analytical evaluations: We agree that component-wise ablation would be an interesting addition to the paper. While our appendix already included ablations on staging and pruning, we now added loss-component ablations to isolate the roles of orthogonality, independence, and cross-modal MI as well as the weighting. The table is included in the update Appendix.
> In summary, we see that for modality-specific labels, the orthogonality loss alone is sufficient to encourage decoupling, since here each modality’s signal lies on its own manifold, so ensuring linear disjointness prevents interference without needing additional statistical constraints.
>
> In contrast, the shared categorical label relies on both orthogonality and independence, because it emerges from the joint statistical structure across modalities; orthogonality separates the spaces geometrically, while independence removes nonlinear correlations and redundancy, allowing the shared subspace to capture only the truly cross-modal information rather than correlated modality-specific noise.
>
> Finally, Mutual-information (MI) retention loss preserves content. Without MI, linear-probe and few-shot performance drops across all subspaces (shared and private).
>
> 3. Potential inconsistencies (Q11): We would like to clarify that there is no contradiction between the loss formulation and the observed improvement in unimodal representations. The independence loss is specifically designed to prevent information leakage between the shared and private subspaces, not to block cross-modal learning altogether. During fine-tuning, each modality benefits from the shared subspace, which captures features that are mutually inferable across modalities. Thus, while the independence term isolates modality-unique content, the shared component still facilitates cross-modal transfer, allowing unimodal representations to become richer and more semantically aligned through joint training.

---

> > ### Author Response · Authors · 2025-11-19
> > **Rebuttal response II**
> >
> > 4. Practical limitations in real-world settings (Q8). We would like to clarify that our method does not require prior knowledge of which modality is dominant or more informative. The fine-tuning is performed in a fully unsupervised manner, without using any task labels or modality weights. The information about which modality is “dominant” is used only for validation, to verify whether the learned subspaces have correctly localized task-relevant information into the appropriate shared or modality-specific components. To assess this, we train a simple classifier (e.g., logistic regression) on each representation component for a given label, whichever component yields the highest predictive accuracy indicates where that information is encoded.
> >
> > Our primary goal in this work is to promote interpretability in multimodal representations. By explicitly disentangling the shared and private subspaces, the framework enables knowledge discovery, it reveals what information each modality uniquely contributes and what information is redundantly shared. In real-world settings this disentanglement can provide actionable insight. For example, in healthcare applications, it can highlight which clinical modality (e.g., imaging, audio, or physiological signals) carries unique diagnostic value, thereby informing which measurements are most critical for improving predictive power or reducing acquisition cost. Finally, Table 3a focuses on the fusion aspect. The “fused” representation corresponds to the concatenation of the fine-tuned subspaces.
> >
> > 5. Unclear experimental design and training procedure (Q3, Q4, Q6, Q9): We would like to clarify that the model is not fine-tuned on any supervised downstream task. Instead, MultiLoReFT performs an unsupervised fine-tuning stage in which fusion and disentanglement are jointly optimized to produce representations that generalize across downstream tasks without further adaptation. The training losses remain consistent across stages, with adaptive reweighting as detailed in Algorithm 1.
> >
> > We describe the convergence criterion in the paper as a minimum relative improvement of 0.001 in the validation loss within a patience window of 40 epochs (extended to 100 epochs for the joint stage to account for recovery after pruning). Meaning that if the validation loss doesn't improve more than 0.001 percent for 40 epochs, training proceeds to the next stage.
> >
> > Regarding the cross-modal mutual information (MI) loss, its purpose is to encourage the shared and modality-specific projections to retain information from their original pretrained embeddings. This ensures that the disentanglement process does not erase valuable unimodal knowledge and that both shared and private subspaces remain informative.
> >
> > 6. Restricted evaluation to two modalities (Q7): Evaluating beyond two modalities is an interesting extension. However, we note that most existing multimodal disentanglement and PID-based frameworks are also restricted to bimodal setups due to the same underlying challenge that defines meaningful notions of “shared” and “modality-specific” information when higher-order interactions emerge.
> > In theory, MultiLoReFT can be extended to multiple modalities by defining a set of projection subspaces that are the combination of shared information and expanding the independence constraints across all relevant component pairs. However, this added granularity could also reduce interpretability as it becomes unclear what constitutes “shared information between two modalities but not a third,” or how such partial factors would be utilized in downstream tasks. We therefore focus on the bimodal case for consistency with prior work and lack of ground truth labels for validation, but note that the formulation is conceptually extensible. We have added this to the updated discussion.
> >
> > 7. Limited analysis of dependence on encoder quality (Q10): We agree that sensitivity to encoder quality is important. In response, we replicated the CREMA-D experiments with stronger unimodal backbones, i.e. the MViT-V2-S (K400)[1] for video and WavLM-Base+[2] for audio. These results are added to the updated manuscript, showing increased performance compared to the Wav2Vec2.0+3D-ResNet18 setting, reaching 80 percent accuracy, comparable to SOTA approaches. This behavior aligns with our goal that MultiLoReFT can directly leverage stronger unimodal encoders to yield better, and more interpretable, multimodal representations, even when multimodal labeled data are limited.
> >
> > [1] Li, Yanghao, et al. "Mvitv2: Improved multiscale vision transformers for classification and detection." Proceedings of the IEEE/CVF conference on computer vision and pattern recognition. 2022.
> >
> > [2] Chen, Sanyuan, et al. "Wavlm: Large-scale self-supervised pre-training for full stack speech processing." IEEE Journal of Selected Topics in Signal Processing 16.6 (2022): 1505-1518.

---

> > > ### Author Response · Authors · 2025-11-19
> > > **Rebuttal response III**
> > >
> > > 8. Fairness of comparisons: We thank the reviewer for raising this important point. Ensuring the fairness and consistency of comparisons across baselines was carefully considered in our experimental design, and we will make this more explicit in the revision. All baselines (DRIM-U, APOLLO, contrastive, …) were trained using the same pretrained unimodal embeddings as input. The only architectural differences lie in the adapters specific to each method: for instance, DRIM-U employs a discriminator-based decoupling mechanism, whereas APOLLO learns latent parameters directly. Also, to support transparency, we include a summary table in the Appendix detailing the number of trainable parameters for each baseline and their adapter configurations.
> > >
> > > Regarding clarification of the unsupervised variant of DRIM-U, we used the unsupervised variant as described in the original paper, which is most aligned with our fully unsupervised fine-tuning framework. The supervised variant requires labeled supervision that is unavailable in our setting. This adaptation does not introduce additional supervision or data imbalance, it simply replaces the supervised loss with its unsupervised formulation as defined by the authors.
> > >
> > >
> > > 9. Limited contextualization within related multimodal literature (Q2): We  agree that PID-based methods are highly relevant to our work, which is why we have a section in the Related Work  with a dedicated paragraph on “Disentangled Multimodal Representations”, discussing these approaches. We have updated the manuscript to better describe MultiLoReFT in the context of such approaches and below is a short summary:
> > > Conceptually, FactorCL and related PID-motivated methods aim to decompose information with respect to a reference variable (typically a task label Y), separating what each modality contributes uniquely or redundantly toward that target. In contrast, MultiLoReFT decouples information independent of the task and in an unsupervised manner. CoMM adopts PID as a theoretical perspective but does not explicitly decompose the learned representation into distinct components. Instead, it maximizes mutual information between augmented multimodal embeddings, encouraging the joint representation to capture redundant, unique, and synergistic information implicitly.

---

### Author Response · Authors · 2025-11-28
**Rebuttal summary**

We would like to thank all reviewers for their thoughtful feedback and constructive suggestions. We have incorporated all of these comments into the revised manuscript, which we believe has substantially improved the clarity and strength of the work. In particular, we have (i) integrated requested analyses, (ii) clarified a few key misunderstandings, and (iii) provided additional theoretical and empirical justification for our design choices. The main points of our rebuttal and revision are as follows:

1. Extended ablations and theoretical justification: As suggested, we expanded the ablation study to systematically examine the impact of each design choice in MultiLoReFT, including individual loss components and training settings. We also added theoretical justification to motivate these choices, aiming to make the method’s behavior more intuitive and its design more clearly justified.

2. Scaling MultiLoReFT and multimodal performance: To demonstrate how MultiLoReFT scales to more complex encoders and affects multimodal performance, we added experiments on the CREMA-D dataset using larger video and audio encoders. These results show that MultiLoReFT can leverage stronger unimodal backbones to achieve competitive, and in some cases SOTA, multimodal performance.

3. Clarification on interpreting shared vs. modality-specific results: A concern raised by one reviewer was that MultiLoReFT does not always outperform all baselines on modality-specific prediction, despite strong shared representation results. We clarify that strong modality-specific performance is only meaningful when it is built on a strong shared representation. A model can “cheat” by pushing most information about a modality into the modality-specific branch and leaving the shared space underutilized (not capturing the meaningful shared information). This yields high modality-specific scores but poor shared performance, indicating weak decoupling rather than true disentanglement. In our view, a method must first learn a good shared representation; only then do gains in modality-specific branches indicate meaningful, well-structured separation of shared and private information.The revised text clarifies this point and explains why large gaps in the baselines do not necessarily indicate meaningful decoupling of shared and private information.

4. Clarified scope and primary contributions: Finally, we clarified the scope and goals of the paper. The main objective is to show that MultiLoReFT can decouple shared and modality-specific representations by fine-tuning pretrained unimodal encoders. Demonstrating that these decoupled representations also support strong multimodal performance serves as a proof of concept, rather than the primary contribution. We have updated the introduction, methodology, and results sections to make clear that our focus is on disentangling meaningful shared vs. modality-specific information, not on exhaustive benchmarking of multimodal fusion architectures.


We hope these revisions address the reviewers’ concerns and make the contributions and positioning of MultiLoReFT clearer.

---

### Meta-Review · Area_Chair_s6Ko · 2026-01-10

**Summary:**

Across four reviews, the scores are 2, 2, 4, and 4 (average 3). The main concerns raised by the reviewers include:
- insufficient evaluations, including limited downstream task evaluation and small gains, sparse ablations on loss components and weighting, and missing comparisons to MI/PID methods (e.g., FactorCL, CoMM).
- technique issues, including restricted bimodal scope and unclear scalability.

**Reviewer Concerns:**

The rebuttal provides theoretical justification for HSIC–orthogonality–MI, introduces loss component ablations, clarifies the unsupervised training pipeline and convergence, specifies fused representations, and reports improved CREMA D accuracy with stronger encoders.

The feedback partially addresses clarity and methodology, but the core concerns about broader baselines, multi-modality scaling, and substantive downstream utility remain only partly resolved.

**Reviewer Scores:**

Reviewer JLeH has decided to increase his score from 4 to 6. However, the other reviewers (including Reviewer NKz5, iJwS, and pWRG) are likely to maintain their scores or only slightly increase them, as some issues remain unresolved.

---

### Decision · Program_Chairs · 2026-01-26

Reject